# Structure of a transcribing Pol II-DSIF-SPT6-U1 snRNP complex

Luojia Zhang ®[1], Christopher Batters ®[1], Shintaro Aibara[2], Yuliya Gordiyenko[1], Kristina Žumer ®[2], Jana Schmitzová[2], Kerstin Maier[2], Patrick Cramer[2] & Suyang Zhang ®[1]✉

In eukaryotic cells, splicing occurs predominantly co-transcriptionally, enhancing splicing efficiency and fidelity while introducing an additional layer of regulation over gene expression. RNA polymerase II (Pol II) facilitates co-transcriptional splicing by recruiting the U1 small nuclear ribonucleoprotein particle (U1 snRNP) to the nascent transcripts. Here, we report the cryo-electron microscopy structure of a transcribing Pol II-U1 snRNP complex with elongation factors DSIF and SPT6. In addition, our biochemical analysis reveals that the phosphorylated Pol II carboxyl-terminal domain and SPT6 interact directly with U1 snRNP proteins, facilitating its recruitment to the elongation complex. This multivalent interaction between U1 snRNP and the transcription elongation complex may both allow efficient spliceosome assembly and ensure transcription processivity.

In eukaryotes, RNA polymerase II (Pol II) synthesizes the precursor messenger RNA (pre-mRNA), which undergoes several processing steps, including splicing, before serving as a template for protein synthesis. During splicing, the spliceosome removes introns from the pre-mRNA while it is being transcribed by Pol II[1–5]. The coupling between transcription and splicing enhances the efficiency and accuracy of splicing. This is exemplified in metazoan genes which often harbour introns that span several kilobases in length[6], raising the intriguing question of how distant intron ends are brought into proximity to ensure precise and efficient splicing. The highly repetitive carboxyl-terminal domain (CTD) of Pol II has been proposed to function as a platform to recruit splicing factors[7–11], yet the Pol II CTD itself is insufficient to stimulate splicing[12].

A direct interaction between Pol II and U1 snRNP, independent of the Pol II CTD, was revealed by a cryo-electron microscopy (cryo-EM) structure[13]. U1 snRNP is the first spliceosome component recruited to the pre-mRNA to recognize the 5′ splice site (5′ SS)[14]. The direct Pol II-U1 snRNP interaction retains the 5′ SS near the RNA exit site of Pol II, thereby bridging the 5′ SS to the branch point sequence and the 3′ splice site as they emerge from Pol II. This intron loop model is further supported by CRISPR interference experiments revealing that the nascent 5′ SS base paired with U1 snRNA is tethered to Pol II during intron synthesis[15].

Nevertheless, it remains unknown how transcription elongation factors affect U1 snRNP recruitment to an elongating Pol II. Release of Pol II into productive elongation requires the positive transcription elongation factor b (P-TEFb)[16,17] which phosphorylates the Pol II CTD, the DRB sensitivity inducing factor (DSIF, a heterodimer of SPT4 and SPT5) and the negative elongation factor (NELF)[18–21]. This change in phosphorylation state results in dissociation of NELF and recruitment of elongation factors SPT6 and PAF[19,22]. SPT6 is both a transcription elongation factor and a histone chaperone, playing an essential role in transcription processivity, elongation across nucleosome barriers and nucleosome positioning[23–26]. Here we report the cryo-EM structure of a transcribing Pol II-DSIF-SPT6-U1 snRNP complex. Additionally, we identify that U1 snRNP is recruited to the elongation complex through multiple interaction sites, enabling efficient co-transcriptional spliceosome assembly.

## Results

### SPT6 and phosphorylated Pol II CTD enhance U1 snRNP binding to Pol II

Human U1 snRNP consists of U1 snRNA, seven Sm proteins and three U1-specific proteins, U1-70K, U1A and U1C. Both U1-70K and U1A contain RNA recognition motifs (RRMs) and associate with the U1 snRNA

---

[1]MRC Laboratory of Molecular Biology, Cambridge, UK. [2]Max Planck Institute for Multidisciplinary Sciences, Department of Molecular Biology, Göttingen, Germany. ✉e-mail: szhang@mrc-lmb.cam.ac.uk

stem loops, while U1C comprises a zinc finger domain that facilitates 5′ SS recognition[27] (Fig. 1a). We first investigated the effect of P-TEFb phosphorylation on the Pol II-U1 snRNP interaction. We generated a CRISPR-knockin cell line with an N-terminal Twin-Strep tag on the largest subunit of Pol II, RPB1, and purified human Pol II (Methods). Using the Twin-Strep tag on Pol II, we performed pulldown experiments in the absence of any nucleic acids to capture solely protein-mediated interactions (Fig. 1b, c). Consistent with the Pol II-U1 snRNP structure[13], U1 snRNP bound directly to Pol II and its binding increased with Pol II phosphorylation by P-TEFb (Fig. 1b compare lanes 10 and 11, 1c). This enhanced binding is attributed to the specific interaction of U1 snRNP with the P-TEFb phosphorylated Pol II CTD, but not with the non-phosphorylated CTD (Fig. 1d). Pulldown experiments using U1-specific proteins revealed that the phosphorylated Pol II CTD bound specifically to the U1-70K[RRM] domain (Fig. 1d, e), the same domain that mediates the direct Pol II-U1 snRNP interaction[13]. We used the RRM domain of U1-70K in the pulldown as full-length U1-70K is insoluble.

To assess the effect of elongation factors on the Pol II-U1 snRNP interaction, we added DSIF and SPT6 to Pol II and treated them with P-TEFb phosphorylation before incubating with U1 snRNP. In the absence of nucleic acids, DSIF failed to associate with Pol II, whereas SPT6 formed a stochiometric complex with Pol II (Fig. 1b lane 12). Interestingly, association of U1 snRNP to phosphorylated Pol II is further enhanced by SPT6 (Fig. 1b compare lanes 11 and 12, 1c), suggesting that SPT6 may play a role in U1 snRNP recruitment and thereby facilitating co-transcriptional splicing.

## Structure of the EC-DSIF-SPT6-U1 snRNP complex

To gain molecular insights into U1 snRNP recruitment during transcription elongation, we assembled a mammalian transcription elongation complex (EC) with human DSIF and SPT6 on a DNA-RNA scaffold. The scaffold contains a DNA mismatch bubble that enables formation of a 9-base pair DNA-RNA hybrid duplex, and a modified MINX pre-mRNA that comprises a 5′ exon and a truncated intron of 34 nucleotides (Fig. 2a). The assembled EC was incubated with DSIF and SPT6, phosphorylated by P-TEFb, and purified by size exclusion chromatography before its incubation with purified human U1 snRNP and subjected to single particle cryo-EM analysis (Fig. 2b, Supplementary Figs. 1–3). Although PAF does not interfere with U1 snRNP binding to Pol II in vitro[13], we omitted PAF in the assembly because it further increased the flexibility of an already highly mobile complex.

We obtained a cryo-EM reconstruction of the EC-DSIF-SPT6-U1 snRNP complex at an overall resolution of 3.5 Å (Fig. 2c, Supplementary Figs. 1–3, Table 1, Supplementary Movie 1). Both U1 snRNP and SPT6 are highly mobile on the Pol II surface. Focused refined maps with a local resolution of 7.3 Å for U1 snRNP and 6.2 Å for SPT6 allowed confident docking of previous models[13,28] (Supplementary Fig. 3). While both SPT6 and DSIF engage the stalk domain of Pol II, DSIF additionally interacts with the upstream DNA, possibly explaining the lack of DSIF binding to Pol II in the absence of nucleic acids in our pulldown experiment (Fig. 1b lane 12). The direct Pol II-U1 snRNP interface is mediated by the RRM domain of U1-70K which contacts the RPB2 and RPB12 subunits of Pol II, consistent with the Pol II-U1 snRNP structure[13]. In addition, similar movements of the U1 snRNA stem loops relative to the U1 snRNP crystal structure[29] were observed (Supplementary Fig. 4a, b). When superimposed on the Sm-ring, the 5′SS-U1 snRNA duplex moves away from the Sm-ring, leaving a larger gap between itself and the Sm-ring, possibly leading to U1C dissociation and explaining the lack of U1C densities in our reconstruction. Overall, our structure reveals that the U1 snRNP interaction with Pol II is

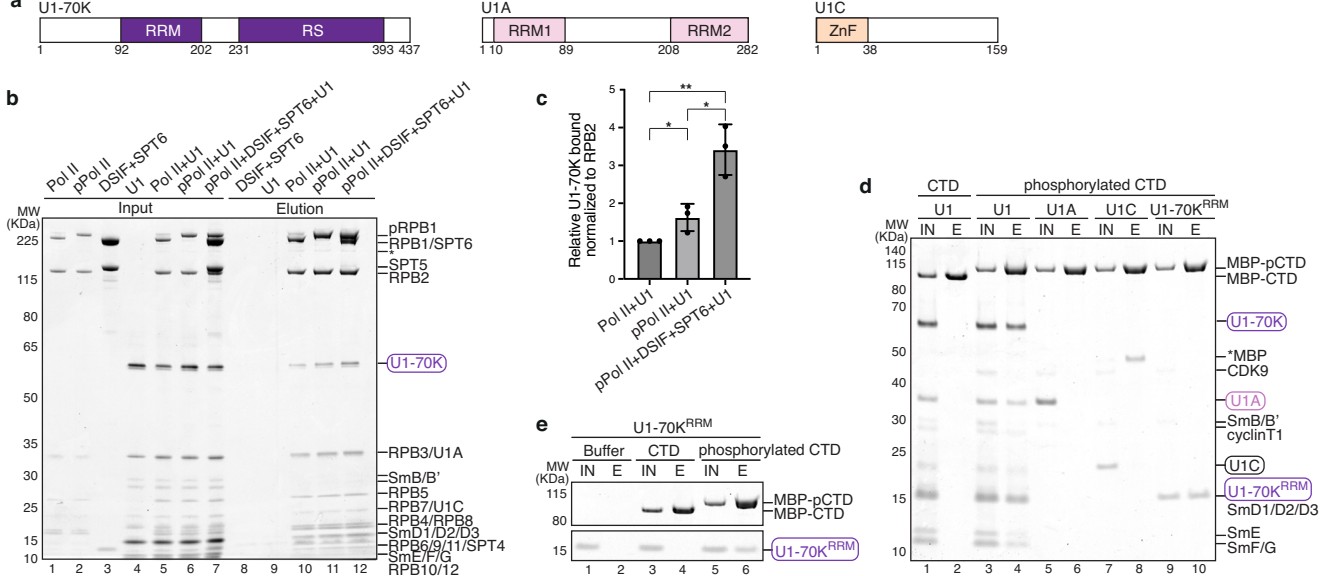

**Fig. 1 | Pol II CTD and SPT6 enhance U1 snRNP binding to Pol II. a** Domain organization of U1-70K, U1A and U1C. RRM: RNA recognition motif, RS: arginine/serine rich domain, ZnF: zinc finger. **b** Pulldown of U1 snRNP (U1) using Twin-Strep-tagged human Pol II in different phosphorylation states, along with elongation factors DSIF and SPT6. The prey protein U1-70K as a part of U1 snRNP is highlighted in a purple oval rectangle. The pulldown was performed in the absence of any nucleic acids. pPol II: phosphorylated Pol II, pRPB1: phosphorylated RPB1. Asterisk indicates RPB1 degradation. **c** Quantification of the amount of U1-70K in the elution which is normalized to respective RPB2 band in each condition and against the Pol II + U1 condition. Data are presented as mean values ± standard deviation. Significance was calculated using two-tailed unpaired Student's t-test from three independent biological replicates. The *P* value for pPol II + U1 versus Pol II + U1 is 0.0392, for pPol II + DSIF + SPT6 + U1 versus pPol II + U1 is 0.015, and for pPol II + DSIF + SPT6 + U1 versus Pol II + U1 is 0.0033. * and ** indicate P ≤ 0.05 and P ≤ 0.01, respectively. **d** Pulldown of U1 snRNP and U1-specific proteins with MBP-tagged human Pol II CTD. U1 snRNP interacts specifically with the P-TEFb phosphorylated Pol II CTD (pCTD). P-TEFb consists of CDK9 and cyclin T1, a truncated cyclin T1 (1-272) was used. The prey proteins U1-70K[RRM], U1A, and U1C are highlighted in coloured rectangles. IN input, E elution, MBP maltose binding protein. Asterisk indicates MBP-contamination from the U1C preparation that was enriched in the elution. **e** Pulldown of U1-70K[RRM] with non-phosphorylated and phosphorylated human Pol II CTD. All pulldowns in the figure were repeated in triplicate. Source data are provided as a Source Data file.

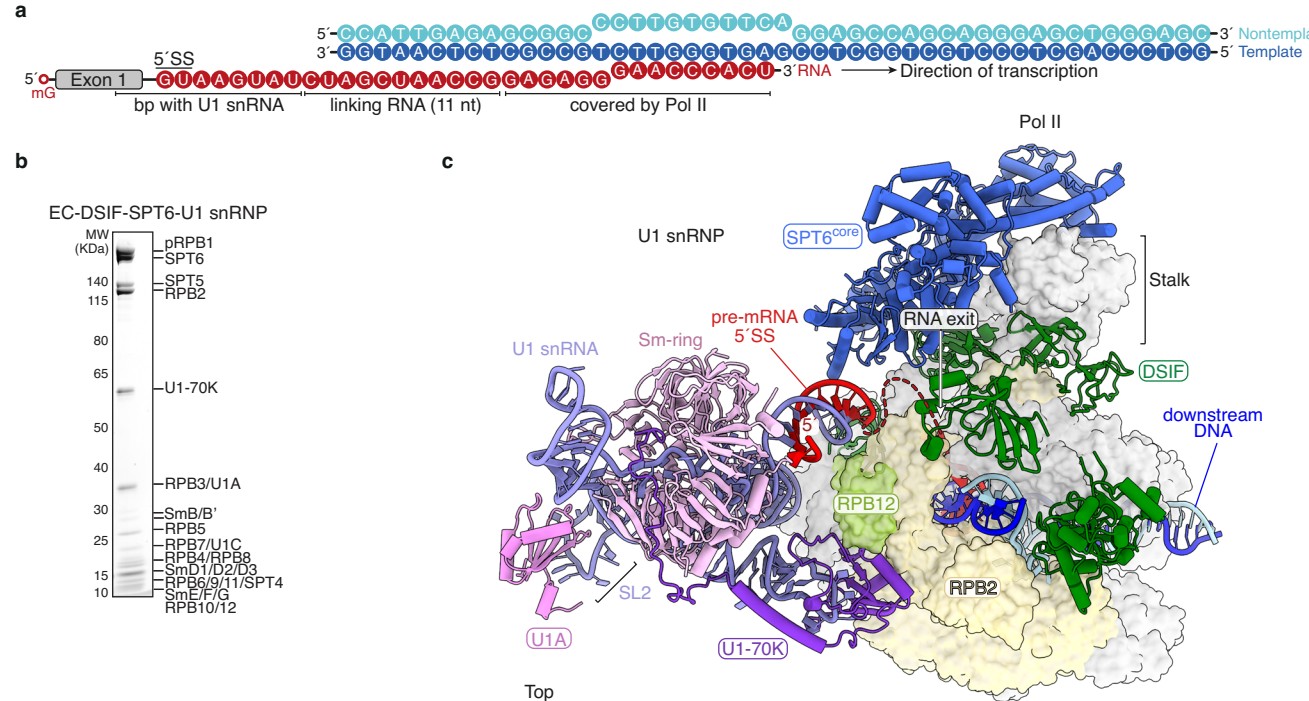

**Fig. 2 | Structure of the EC-DSIF-SPT6-U1 snRNP complex. a** Nucleic acid scaffold used for cryo-EM analysis of the EC-DSIF-SPT6-U1 snRNP complex, with template DNA in dark blue, non-template DNA in cyan and RNA in red. mG: 7-methyl guanosine, bp base pair. **b** Assembled EC-DSIF-SPT6-U1 snRNP complex on a 4–12% NuPAGE Bis-Tris gel, run in MOPS, stained with Instant Blue. The complex assembly experiment was repeated in triplicate. Source data are provided as a Source Data file. **c** Structure of the EC-DSIF-SPT6-U1 snRNP complex in top view. Pol II is shown in light grey surface representation except for RPB2 in yellow and RPB12 in lime. Elongation factors SPT6 (blue) and DSIF (green) as well as U1 snRNP are shown in cartoon representation. The backbone of U1 snRNA is colored in lavender, U1-70K in dark purple, U1A in violet and Sm proteins in light pink. Template DNA is colored dark blue, non-template DNA in cyan and RNA in red. SL stem loop.

compatible with the presence of elongation factors DSIF and SPT6 on the Pol II surface. However, we could not resolve the interactions of U1 snRNP with SPT6 and the phosphorylated Pol II CTD in our cryo-EM reconstruction, likely due to the mobile nature of the complex.

### SPT6 interacts directly with U1A

To understand the role of SPT6 in U1 snRNP recruitment, we performed pulldown experiments using an MBP-tagged SPT6 and observed direct binding of U1 snRNP to SPT6 regardless of its phosphorylation state (Fig. 3a, b). Human SPT6 contains a core domain that engages the Pol II stalk domain (Fig. 2c) and a tSH2 domain that interacts with the phosphorylated Pol II CTD linker region[19,22]. The SPT6 core and tSH2 domains are flanked by an N-terminal domain (NTD) and a C-terminal region (CTR) that are only present in eukaryotes and are less well characterized (Fig. 3a). SPT6[NTD], a highly acidic and largely unstructured region, is important for its interaction with histones, while SPT6[CTR] is mostly disordered. Deletion of the CTR (SPT6[ΔCTR]) completely abolished U1 snRNP binding (Fig. 3b lane 10), whereas eliminating either both the core and tSH2 domains (SPT6[NTD+CTR]) or the NTD (SPT6[ΔNTD]) did not affect U1 snRNP association (Fig. 3c lanes 6 and 10). Consistently, while SPT6[CTR] itself is able to bind U1 snRNP (Fig. 3b lane 12), neither SPT6[Core] nor SPT6[NTD] interacts with U1 snRNP (Fig. 3b, c lanes 8), underlying the importance of SPT6[CTR] in mediating the SPT6-U1 snRNP interaction.

We next investigated which U1-specific proteins are responsible for interacting with SPT6[CTR] and found that SPT6[CTR] bound specifically to U1A, but not U1C and U1-70K[RRM] (Fig. 3d). We further tested U1A interaction with individual SPT6 truncations (Fig. 3a, e). Consistent with the SPT6-U1 snRNP interaction, deletion of the CTR (SPT6[ΔCTR]) eliminated binding of U1A, while the CTR is the only domain capable to engage U1A by itself (Fig. 3e lanes 10 and 8). This data was further supported by isothermal titration calorimetry (ITC) experiments,

obtaining a dissociation constant (Kd) of 1.96 μM for U1A interaction with wildtype SPT6 and a Kd of 4.02 μM for U1A interaction with SPT6[CTR] (Fig. 3a, Supplementary Fig. 5). The increased affinity of SPT6[NTD+CTR] to U1A (Kd of 1.34 μM) is likely contributed by the strong acidic nature of SPT6[NTD] which mimics RNA substrates of RRM domains.

U1A associates with stem loop 2 of U1 snRNA, located distant from the Pol II-U1 snRNP interface (Fig. 2c). The dynamic nature of both U1 snRNP and SPT6 on Pol II makes it challenging to resolve their interactions by cryo-EM. Additionally, AlphaFold3 (ref. 30) was unable to generate confident predictions when full-length sequences of SPT6 and U1A were used. However, guided by the SPT6[CTR]-U1A interaction from our pulldown and ITC experiments, AlphaFold3 produced a prediction for SPT6[CTR] bound to U1A with high confidence (Fig. 4a). A C-terminal α-helix of SPT6 (residues 1671-1696) harbouring aromatic and hydrophobic residues contacts a hydrophobic surface of the N-terminal RRM domain of U1A, located on the opposite side of its U1 snRNA interface. Disruption of the SPT6[CTR]-U1A interaction by replacing the SPT6 C-terminal helix with GSA linkers (SPT6[Δhelix]) abolished U1A binding, while mutation of large aromatic and hydrophobic residues at the interface (SPT6[3A]: W1674A, W1681A, L1682A) strongly reduced the SPT6-U1A interaction (Fig. 4b). This result validates the predicted interaction of the C-terminal helix of SPT6 with U1A.

## Discussion

Taken together, our results reveal that U1 snRNP is recruited to the transcription elongation complex through three contact points: while the Pol II body and the phosphorylated Pol II CTD interact directly with U1-70K, the elongation factor SPT6 associates with U1A (Fig. 4c, Supplementary Movie 1). All three interactions are mediated by the RRM domains in U1 snRNP. Following U1 snRNP recruitment, U2 snRNP joins to recognize the branch point sequence, forming the pre-spliceosome.

**Table 1 | Statistics of cryo-EM reconstructions and structural model**

| | Overall (EMD-53087) | SPT6 (EMD-53088) | U1 snRNP (EMD-53089) |
|---|---|---|---|
| **Data collection and processing** | | | |
| Microscope | Titan Krios | | |
| Voltage (kV) | 300 | | |
| Camera | K3 | | |
| Magnification | 81,000 x | | |
| Pixel size (Å/pixel) | 1.05 | | |
| Electron exposure (e⁻/Å²) | 40.4 | | |
| Exposure rate (e⁻/Å²/frame) | 1.01 | | |
| Number of frames per movie | 40 | | |
| Defocus range (μm) | 0.5–2.0 | | |
| Automation software | SerialEM | | |
| Symmetry imposed | C1 | | |
| Initial particle numbers | 2,039,129 | | |
| Final particle numbers | 52,065 | | |
| Map sharpening $B$ factor (Å²) | −111.322 | −500 | −513.8 |
| Map resolution (Å, FSC = 0.143) | 3.5 | 6.2 | 7.3 |
| **Refinement** | | | |
| Initial models used (PDB) | 7BOY, 9HVQ | | |
| Model resolution (Å) | 3.7 | | |
| Model composition | | | |
| Non-hydrogen atoms | 56,145 | | |
| Protein residues | 6,297 | | |
| Nucleic acid residues | 263 | | |
| Ligands | Zn:8 Mg:1 | | |
| $B$ factors (Å²) | | | |
| Protein | 327.52 | | |
| Nucleotide | 554.97 | | |
| Ligand | 239.11 | | |
| R.m.s. deviations | | | |
| Bond lengths (Å) | 0.004 | | |
| Bond angles (°) | 0.568 | | |
| Validation | | | |
| MolProbity score | 1.57 | | |
| Clash score | 5.98 | | |
| Poor rotamers (%) | 0.00 | | |
| Cβ deviations (%) | 0 | | |
| Ramachandran plot | | | |
| Favored (%) | 96.35 | | |
| Allowed (%) | 3.65 | | |
| Outliers (%) | 0.00 | | |
| PDB code | 9QEQ | | |

Superposition of the yeast pre-spliceosome structure[31] with our EC-DSIF-SPT6-U1 snRNP structure reveals that U2 snRNP can be accommodated on the elongating Pol II (Supplementary Fig. 4c, d), indicating that the pre-spliceosome may assemble co-transcriptionally.

In addition to its role in splicing, U1 snRNP recruitment to the nascent transcripts is required to suppress pre-mature cleavage and polyadenylation from cryptic polyadenylation signals in human cells[32–34]. Furthermore, U1 snRNP was found to stimulate synthesis of long introns by increasing Pol II elongation rate in mammalian cells[35]. In the absence of U1 snRNP, Pol II is more susceptible to termination and arrest, preventing the production of full-length transcripts. A possible explanation is that U1 snRNP may prevent the recruitment or stable engagement of the 3′ end processing machinery to cryptic sites. Moreover, the integrator complex can lead to pre-mature termination and cleavage of transcripts in the promoter-proximal region[36–40]. Superposition of the integrator-containing pre-termination complex[41] with the EC-DSIF-SPT6-U1 snRNP structure reveals that the integrator tail module, which is required to facilitate the release of nucleic acids from the Pol II cleft and transcription termination, clashes with U1 snRNP on the Pol II surface (Supplementary Fig. 6). Furthermore, the integrator core module clashes with the SPT6 core domain. Therefore, U1 snRNP may prevent stable binding of the integrator complex while stabilizing SPT6 on Pol II. Additionally, U1 snRNP may compete with the integrator-associated protein phosphatase 2A[37] for the Pol II CTD to prevent its dephosphorylation and transcription termination.

Overall, the multivalent interactions between the elongating Pol II and U1 snRNP facilitate the swift recruitment of U1 snRNP to the nascent transcripts, which may allow efficient and accurate co-transcriptional spliceosome assembly and ensure transcription processivity.

## Methods

### Cloning and protein expression

Human U1A, U1C, and U1-70K^RRM were amplified from human cDNA and cloned into 1C vector (addgene no. 29654) with an N-terminal His₆-MBP-TEV tag using ligation independent cloning[42]. All SPT6 truncations were generated using ligation independent cloning, Quikchange[43] or NEB HiFi assembly (SPT6^NTD+CTR) into 438 C vector (addgene no. 55220) with an N-terminal His₆-MBP-TEV tag. SPT6^NTD+CTR contains a 3x GSA linker between the NTD and CTR.

Proteins cloned into 438 C vector were expressed in High Five cells (Gibco). 1 L High Five cells in Sf-900™ II SFM medium (Gibco) were infected with P2 virus and grown for 50–72 h. Cells were harvested by centrifugation at 1000xg for 18 min and frozen in liquid nitrogen, and stored at −80 °C before protein purification.

Proteins cloned into 1 C vector were expressed in BL21 (DE3) RIL cells (Agilent) in LB medium. Expression was induced with 1 mM IPTG when the cell density reached an OD of 0.6–0.8. Proteins were expressed overnight at 18 °C before harvesting and storage at −80 °C.

The following primers were used for cloning:

U1A_Fw: TACTTCCAATCCAATGCAATGGCAGTTCCCGAGACCC

U1A_Rv: TTATCCACTTCCAATGTTATTACTACTTCTTGGCAAAGGAGATCTTCATGG

U1C_Fw: TACTTCCAATCCAATGCAATGCCCAAGTTTTATTGTGACTACTGC

U1C_Rv: TTATCCACTTCCAATGTTATTATCTGTCTGGTCGAGTCATTCCG

U1-70K^RRM_Fw: TACTTCCAATCCAATGCACACAATGATCCCAATGCTC

U1-70K^RRM_Rv: TTATCCACTTCCAATGTTATTATCCTCTTCTGGTACCACC

SPT6^NTD_Fw: TACTTCCAATCCAATGCAATGTCTGATTTTGTGGAAAGCGAG

SPT6^NTD_Rv: TTATCCACTTCCAATGTTATTATGTGAGGTGGCTGCTTTCTAGC

SPT6^NTD+CTR_Fw_insert (insertion of CTR with GSA linker): GCTGGTTCCGCTGGTTCCGCCCCAGGCATCACCCCTAGCA

SPT6^NTD+CTR_Rv_insert (insertion of CTR): CCACTTCCAATGTTATTACCGATCCATCTCGTCCAGGA

SPT6^NTD+CTR_Fw_vector (SPT6^NTD as backbone): GAGATGGATCGGTAATAACATTGGAAGTGGATAACGGATCCG

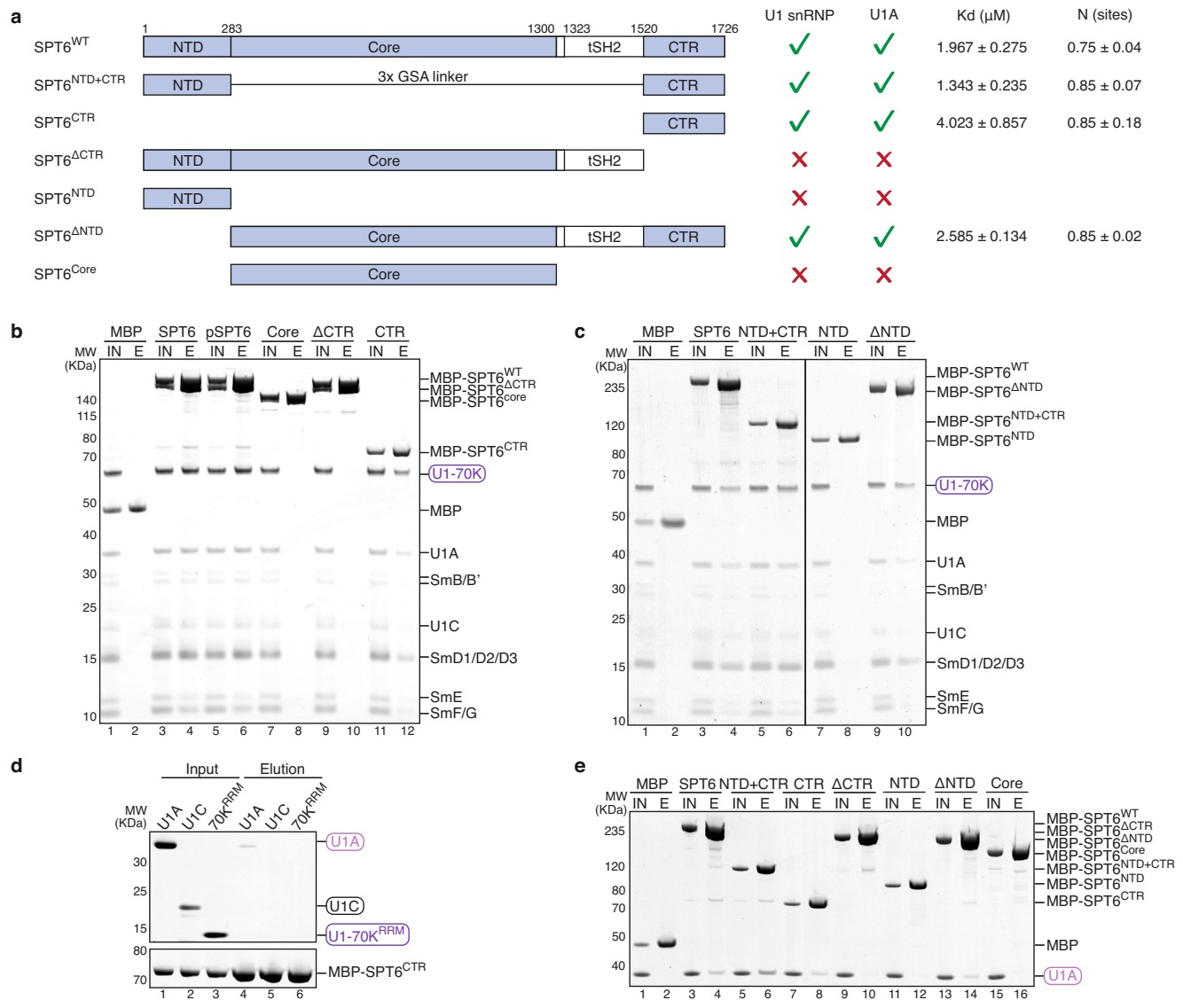

**Fig. 3 | SPT6 interacts directly with U1 snRNP. a** Domain organization of SPT6 and truncated SPT6 constructs. A summary of U1 snRNP and U1A binding to all SPT6 constructs in pulldown assays is shown on the right, together with the dissociation constant (Kd) and number of binding sites (N) determined by ITC for the SPT6-U1A interaction. **b, c** Pulldown of U1 snRNP using different MBP-tagged SPT6 constructs. The prey protein U1-70K as a part of U1 snRNP is highlighted in a purple oval rectangle. MBP is used as a negative control. IN input, E elution, pSPT6 phosphorylated SPT6. **d** Pulldown of U1-specific proteins using MBP-tagged SPT6^CTR. The prey proteins U1A, U1C, and U1-70K^RRM are highlighted in oval rectangles. **e** Pulldown of U1A using different MBP-tagged SPT6 constructs. The prey protein U1A is highlighted in a violet oval rectangle. All pulldowns were repeated in triplicate. Source data are provided as a Source Data file.

---

SPT6^NTD+CTR_Rv_vector (SPT6^NTD as backbone with GSA linker): GGAACCAGCGGAACCAGCGGAACCTGTGAGGTGGCTGCTTTCTAGC

SPT6^CTR_Fw: TACTTCCAATCCAATGCACCAGGCATCACCCCTAGCA

SPT6^CTR_Rv: TTATCCACTTCCAATGTTATTACCGATCCATCTCGTCCAGGA

SPT6^ΔNTD_Fw: TACTTCCAATCCAATGCAGATCAGGACAATGAAATCCGAGCCA

SPT6^ΔNTD_Rv: same as SPT6^CTR_Rv

SPT6^ΔCTR_Fw: same as SPT6^NTD_Fw

SPT6^ΔCTR_Rv: TTATCCACTTCCAATGTTATTATGGTACAGGATCCTGGTAGTGATCC

SPT6^Δhelix_Fw (with 4x GSA linker): AGTCCAACAGCCATGCAGGTTCCGCTGGTTCCGCTGGTTCCGCCGGTTCCGCTACACCTCGGCCCTCCCCC

SPT6^Δhelix_Rv: TGCATGGCTGTTGGACTTTGGCTGCTGCTGCCGTTGC

SPT6^3A_Fw: CCATGCAGCCATCGACGCTGGAAAAATGGCGGAGCAGGCTGCTCAGGAAAAGGAGGCAGAACGGCGG

SPT6^3A_Rv: GTCGATGGCTGCATGGCTGTTGGACTTTGGCTGCTGCTGCC

## Cell line genome editing

The HEK293 cell line was CRISPR/Cas9 edited to incorporate a His$_{10}$-Twin-Strep tag followed by a TEV cleavage site at the N-terminus of RPB1. Cloning and endogenous knock-in utilized a MMEJ-assisted gene knock-in strategy[44] as described previously[45] with minor modifications. In brief, HEK293 cells were plated 24 h before the start of the experiment to ensure exponential growth. For knock-in, 1 million cells per reaction were transfected with 2 μg DNA (microhomology-containing repair template plasmid and sgRNA-Cas9 expression vector) using Amaxa Nucleofector with SF Cell Line 4D-Nucleofector X Kit L (Lonza V4XC-2012) and program CM130. After electroporation, cells were taken up in 500 μl pre-warmed growth medium, transferred to 6-well plates containing 1.5 mL warm growth medium and allowed to recover for a total of four days at 37 °C and sub-cultured to maintain the cells as

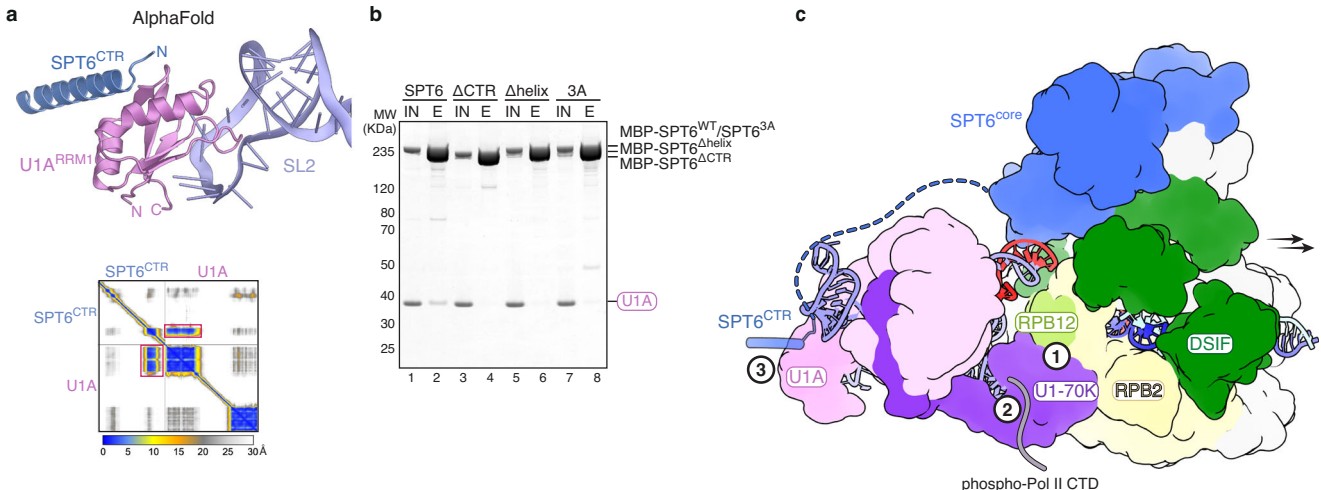

**Fig. 4 | Three contact points facilitate the recruitment of U1 snRNP to the elongating Pol II. a** AlphaFold3 prediction of the SPT6[CTR]-U1A interaction, superimposed onto the U1 snRNP structure[13]. The predicted aligned error plot is shown in the lower panel, with the interface highlighted with red boxes. **b** Mutations that disrupt the predicted SPT6[CTR]-U1A interaction strongly reduced the amount of U1A bound to MBP-tagged SPT6. The SPT6 C-terminal helix was replaced with GSA linkers that can span the length of the helix (SPT6[Δhelix]), or large aromatic and hydrophobic residues in the SPT6 C-terminal helix were mutated to alanines (SPT6[3A]: W1674A, W1681A, L1682A). This pulldown was repeated in triplicate. Source data are provided as a Source Data file. **c** Cartoon schematic showing the three contact points between U1 snRNP and the transcription elongation complex: U1-70K[RRM] with RPB2 and RPB12, U1-70K[RRM] with the phosphorylated Pol II CTD, and U1A with SPT6[CTR]. Arrows indicate the direction of transcription.

needed. On day four, antibiotic selection was initiated by treating with 1 µg/ml puromycin for five days. Presence of the insert in the pool was validated by performing genotyping PCR before proceeding to single cell cloning. Individual clones were isolated by performing serial dilution in 96-well plates and expanding clones for 12–14 days. Obtained clones were characterized by performing genotyping PCR and Sanger sequencing of PCR amplicons of the genomic integration site.

The following primer and insert sequences were used:

sgRNA target sequence: GCCTCCGCCATGCACGGGGG

HR template cloning forward primer: GCGTTACATAGCATCGTACGCGTACGTGTTTGGCCTGCCTCCGCCATGCACGGGACCGAGTACAAGCCCACG

HR template cloning reverse primer: AGCATTCTAGAGCATCGTACGCGTACGTGTTTGGCCCCCGAGGGGGGGGGCACCCCCATGTCCGGACGCGTTTGACTGG

Puro_His10_Twin-Strep sequence: ATGACCGAGTACAAGCCCACGGTGCGCCTCGCCACCCGCGACGACGTCCCCAGGGCCGTACGCACCCTCGCCGCCGCGTTCGCCGACTACCCCGCCACGCGCCACACCGTCGATCCGGACCGCCACATCGAGCGGGTCACCGAGCTGCAAGAACTCTTCCTCACGCGCGTCGGGCTCGACATCGGCAAGGTGTGGGTCGCGGACGACGGCGCCGCGGTGGCGGTCTGGACCACGCCGGAGAGCGTCGAAGCGGGGGCGGTGTTCGCCGAGATCGGCCCGCGCATGGCCGAGTTGAGCGGTTCCCGGCTGGCCGCGCAGCAACAGATGGAAGGCCTCCTGGCGCCGCACCGGCCCAAGGAGCCCGCGTGGTTCCTGGCCACCGTCGGCGTCTCGCCCGACCACCAGGGCAAGGGTCTGGGCAGCGCCGTCGTGCTCCCCGGAGTGGAGGCGGCCGAGCGCGCCGGGGTGCCCGCCTTCCTGGAGACCTCCGCGCCCCGCAACCTCCCCTTCTACGAGCGGCTCGGCTTCACCGTCACCGCCGACGTCGAGGTGCCCGAAGGACCGCGCACCTGGTGCATGACCCGCAAGCCCGGTGCCGGCTCTGGAGCTACTAACTTCAGCCTGCTGAAGCAGGCTGGAGACGTGGAGGAGAACCCTGGACCTCATCATCATCACCACCATCACCATCATCACGGTTCTTCTTGGTCTCACCCCCAATTTGAGAAAGGCGGTGGCAGCGGCGGCGGTAGCGGAGGTGGCAGCTGGTCACACCCACAATTCGAGAAAGGCGCCGAGAATCTTTATTTCCAGTCAAACGCGTCCGGA

Genotyping forward primer: GTAGTGAGGTTTGCGCCTGC

Genotyping reverse primer: GAAAGCGCCAAGTTTCCGCA

## Protein purification

For purification of Twin-Strep-tagged human Pol II, CRISPR/Cas9 edited HEK293 cells were grown in suspension in Expi293 medium (Thermo Fisher) and harvested at a density between 5-6 million/ml. The cell pellet was resuspended in 0 M buffer (50 mM Tris-HCl, pH 7.9, 5 mM MgCl₂, 0.5 mM EDTA, 10 % Glycerol, 2 mM DTT, 1 mM Na₂S₂O₅, 1 mM PMSF) supplemented with EDTA-free protease inhibitor tablets (Roche). The resuspended cells were sonicated before slowly adding an equal volume of 0.6 M ammonium sulfate buffer (0 M buffer with 0.6 M ammonium sulfate) supplemented with EDTA-free protease inhibitor tablets (Roche) and 0.01 mg/ml DNase (Sigma-Aldrich), which was further sonicated. The mixture was cleared by centrifugation, and the supernatant was precipitated with 50% saturated ammonium sulfate for 1 h. The pellet was re-dissolved in 0 M buffer until the conductivity of the sample matched that of the 0.5 M ammonium sulfate buffer, cleared by centrifugation, and applied onto the Strep-TactinXT 4Flow column (IBA). The column was washed with 0.5 M buffer, followed by 0.18 M buffer, and eluted with 0.18 M buffer supplemented with 50 mM biotin. The eluate was loaded onto an UNO Q1 column (BioRad), washed with 0.18 M buffer, and eluted with a linear gradient to 0.5 M buffer. The peak fractions were concentrated using an Amicon Ultra Centrifugal Filter with 100 kDa molecular weight cutoff (MWCO) and buffer exchanged to Pol II storage buffer (20 mM HEPES-NaOH, pH 7.5, 150 mM NaCl, 10 µM ZnCl₂, 2 mM DTT).

Porcine Pol II was purified from *Sus scrofa domesticus* thymus[13]. In brief, the thymus was homogenized, precipitated with polyethylenimine, and purified with Macro-Prep High Q resin (Biorad). The eluate was precipitated with ammonium sulfate, re-dissolved for loading onto an 8WG16 antibody column followed by an Uno Q1 column (Biorad). Peak fractions were concentrated and buffer exchanged to Pol II storage buffer.

Human U1 snRNP was purified from HeLa nuclear extract using immunoaffinity purification and a 10-30% glycerol gradient in 20 mM HEPES-KOH, pH 7.9, 150 mM KCl, 1.5 mM MgCl₂, 1 mM DTT[13]. Fractions containing U1 snRNP were applied onto a Mono Q 5/50 GL column (Cytiva) and eluted with a linear gradient to 1 M NaCl. Peak fractions of U1 snRNP were frozen directly in liquid nitrogen and stored at −80 °C.

For all His$_6$-MBP-tagged human SPT6 constructs, cell pellets were resuspended in SPT6 Lysis Buffer (50 mM Tris-HCl, pH 7.5, 500 mM NaCl, 30 mM Imidazole, 5% glycerol, 1 mM DTT) supplemented with 1 mM PMSF and EDTA-free protease inhibitor tablets (Roche), sonicated, and clarified by centrifugation. Cleared lysates were applied onto the HisTrap HP column (Cytiva), washed with SPT6 Lysis Buffer and Buffer B (50 mM Tris-HCl, pH 7.5, 1 M NaCl, 5% glycerol, 1 mM DTT), followed by elution with SPT6 Lysis Buffer supplemented with 300 mM Imidazole. The eluate was applied onto a home-packed amylose column (NEB), washed with SPT6 Lysis Buffer, and eluted with 50 mM Tris-HCl, pH 7.5, 300 mM NaCl, 50 mM maltose, 5% glycerol, 1 mM DTT. The eluate was diluted to 75 mM NaCl with Buffer A (50 mM Tris-HCl, pH 7.5, 5% glycerol, 1 mM DTT), treated with lambda phosphatase at 1 mM MnCl$_2$ to remove insect cell-derived phosphorylation, and applied onto a HiTrap Heparin HP column (Cytiva). The protein was eluted with a linear gradient to Buffer B, followed by a final size exclusion chromatography (SEC) step on HiLoad 16/600 Superdex 75 pg or 200 pg columns (Cytiva) in SPT6 SEC Buffer (20 mM HEPES-NaOH, pH 7.5, 300 mM NaCl, 5% glycerol, 1 mM DTT). Peak fractions were concentrated, frozen in liquid nitrogen, and stored at −80 °C. Human SPT6 constructs without any tag were purified the same way as His$_6$-MBP-tagged SPT6 with a HisTrap HP column (Cytiva) followed by the amylose column (NEB). Following amylose elution, the tag was removed with TEV protease and proteins were treated with lambda phosphatase at 1 mM MnCl$_2$. Cleaved proteins were re-applied onto the HisTrap HP column to remove the His$_6$-MBP tag and further purified using HiLoad 16/600 Superdex 75 pg or 200 pg columns in SPT6 SEC Buffer.

Human P-TEFb and DSIF were purified as described[22]. In brief, P-TEFb was expressed in High Five cells and purified using GSTrap 4B columns (Cytiva). Following TEV cleavage, proteins were re-applied onto GSTrap 4B columns and further purified using the HiLoad 16/600 Superdex 200 pg column (Cytiva) in protein storage buffer (20 mM HEPES-NaOH, pH 7.5, 300 mM NaCl, 10% glycerol, 1 mM DTT). DSIF was expressed in E. coli and purified by a HisTrap HP column. Following TEV cleavage and application onto the HisTrap HP column, cleaved proteins were loaded onto a HiTrap Q HP column before the final step on the HiLoad 16/600 Superdex 200 pg column in protein storage buffer with 500 mM NaCl.

The His$_6$-MBP-tagged human Pol II CTD was purified as described[28]. The protein was purified with a HisTrap HP column (Cytiva), followed by an amylose column (NEB) and a HiTrapQ HP column. The protein was treated with lambda phosphatase at 1 mM MnCl$_2$ to remove insect cell-derived phosphorylation before applying onto a Superdex 200 increase 10/300 gl column (Cytiva) in SPT6 SEC Buffer.

For U1A, U1C, and U1-70K$^{RRM}$, cell pellets were resuspended in U1A Lysis Buffer (20 mM HEPES-NaOH, pH 7.5, 500 mM NaCl, 10% glycerol, 1 mM DTT) supplemented with 1 mM PMSF and EDTA-free protease inhibitor tablets (Roche). Resuspended cells were sonicated, cleared by centrifugation, and applied onto the HisTrap HP column. The column was washed with U1A Lysis Buffer followed by Buffer D (20 mM HEPES-NaOH, pH 7.5, 1 M NaCl, 10% glycerol, 1 mM DTT) and eluted with U1A Lysis Buffer supplemented with 300 mM Imidazole. The eluate was loaded onto a home-packed amylose column, washed with U1A Lysis Buffer and eluted with 20 mM HEPES-NaOH, pH 7.5, 300 mM NaCl, 100 mM maltose, 10% glycerol and 1 mM DTT. The eluate was diluted to 100 mM NaCl with Buffer C (20 mM HEPES-NaOH, pH 7.5, 10% glycerol and 1 mM DTT) and digested with TEV protease overnight at 4 °C to cleave off the His$_6$-MBP tag. Proteins were applied onto a HiTrap Heparin HP column and eluted with a linear gradient to Buffer D. Peak fractions containing digested proteins were loaded onto a HiLoad 16/600 Superdex 75 pg column in U1A SEC Buffer (20 mM HEPES-NaOH, pH 7.5, 300 mM NaCl, 1 mM DTT). Peak fractions were concentrated, frozen in liquid nitrogen, and stored at −80 °C.

## Pulldown assay

For the pulldown of U1 snRNP with Pol II, Twin-Strep-tagged human Pol II (12 pmol) alone or with SPT6 (36 pmol) and DSIF (36 pmol) was in vitro phosphorylated with P-TEFb (3 pmol) and ATP (1 mM) in SEC100 Buffer (20 mM HEPES-NaOH, pH 7.5, 100 mM NaCl, 3 mM MgCl$_2$ and 1 mM DTT) for 30 min at 30 °C. Protein storage buffer is used instead of P-TEFb as a negative control for the non-phosphorylated Pol II condition. Samples were incubated with U1 snRNP (36 pmol) on ice for 30 min, followed by incubation with the Strep-TactinXT 4Flow high-capacity resin (IBA) equilibrated in K75 Buffer (20 mM HEPES-KOH, pH 7.5, 75 mM KCl, 3 mM MgCl$_2$ and 1 mM DTT) at 4 °C for 1 h. The resin was washed with K75 Buffer and eluted with K75 Buffer supplemented with 50 mM biotin. The eluate was separated on a 4-12% NuPAGE Bis-Tris gel and stained with InstantBlue (Abcam). The amount of U1-70K in the elution was quantified in ImageJ[46] and normalized to respective RPB2 band in each condition and against the Pol II + U1 snRNP condition and plotted using Prism 10 (Graphpad). A two-tailed Student's t-test was used to calculate the significance. One asterisk indicates significance smaller than 0.05, and two asterisks indicate significance smaller than 0.01. The experiment was repeated in triplicate, and all measurements were taken from independent biological replicates. All values in the graph are represented as mean ± standard deviation.

For pulldowns of U1 snRNP or U1-specific proteins with Pol II CTD, His$_6$-MBP-tagged Pol II CTD (50 pmol) was in vitro phosphorylated with P-TEFb (12.5 pmol) and ATP (1 mM) in SEC100 Buffer for 30 min at 30 °C. Protein storage buffer was used instead of P-TEFb as a negative control for the non-phosphorylated CTD condition. The CTD was incubated with U1 snRNP (50 pmol), U1A (100 pmol), U1C (100 pmol) or U1-70K$^{RRM}$ (100 pmol) on ice for 30 min, followed by incubation with amylose resin equilibrated in G-SEC150 Buffer (20 mM HEPES-NaOH, pH 7.5, 150 mM NaCl, 3 mM MgCl$_2$, 5% glycerol and 1 mM DTT) at 4 °C for 1 hour. The resin was washed with G-SEC150 Buffer and eluted with G-SEC150 Buffer supplemented with 20 mM maltose.

For pulldowns of U1 snRNP with different SPT6 constructs, equimolar U1 snRNP (30 pmol) was incubated with His$_6$-MBP-tagged SPT6 proteins (30 pmol) at room temperature for 30 min in SEC100 Buffer in Fig. 3b or SEC200 Buffer (SEC100 with 200 mM NaCl) in Fig. 3c based on the stability of the SPT6 constructs. The mixture was incubated with amylose resin (NEB) at 4 °C for 1 h. The resin was washed with SEC100/200 Buffer and eluted with SEC100/200 Buffer supplemented with 20 mM maltose.

Pulldowns of U1A with different SPT6 constructs were performed as above in SEC200 Buffer except that 3x molar excess of U1A (300 pmol) in Fig. 3e and 5x molar excess of U1A (500 pmol) in Fig. 4b was incubated with His$_6$-MBP-tagged SPT6 constructs (100 pmol). For the pulldown of SPT6$^{CTR}$ with U1-specific proteins, His$_6$-MBP-tagged SPT6$^{CTR}$ (200 pmol) was incubated with 4x molar excess of U1A, U1C, U1-70K$^{RRM}$ (800 pmol). The pulldown was performed as above in SEC100 Buffer.

## ITC

Affinities between SPT6 constructs and U1A were determined by ITC at 25 °C with a Malvern Panalytical ITC200 instrument in SEC300 Buffer (20 mM HEPES-NaOH, pH 7.5, 300 mM NaCl). In a typical ITC experiment, U1A was loaded into a 40 µl syringe at a concentration between 150-550 µM and the SPT6 construct was placed into the sample cell at a concentration between 10 and 35 µM. Titrations consisted of 19 injections of 2 µL preceded by a small 0.5 µL pre-injection that was not used during curve fitting. Experiments were performed at a reference power of 6 µcal/s and an initial delay of 180 s with injections at 180 s intervals with constant stirring at 750 rpm. Control measurements of injections of protein into buffer were performed, and these control heats were close to the values seen for buffer into buffer blank runs. All ITC binding data were corrected with the appropriate control heats of

dilution and fitted using the one set of binding sites' model in Malvern Panalytical PEAQ-ITC analysis software (v1.41) and plotted in Prism 10 (Graphpad). Experiments were performed at least 3 times with different batches of proteins.

## RNA preparation
The following RNA construct was generated for this study:

5′-CUU GGA UCG GAA ACC CGU CGG CCU CCG A**CA GGU AAG UAU** AUG UAU AAC CG*G AGA GGG AAC CCA CU*-3′

The sequence that is complementary to U1 snRNA is in bold, and the sequence that is covered within Pol II is in italic. The construct was cloned into the pUC18 vector with a T7 promotor at the 5′ end and a hepatitis delta virus ribozyme at the 3′ end. RNA was prepared by T7 transcription, followed by purification by gel extraction and capped with Vaccinia capping enzyme as described[13].

## Sample preparation for cryo-EM
The EC-DSIF-SPT6-U1 snRNP complex was formed on a DNA scaffold with the following sequences: template DNA 5′-GCT CCC AGC TCC CTG CTG GCT CCG AGT GGG TTC TGC CGC TCT CAA TGG-3′, non-template DNA 5′-CCA TTG AGA GCG GCC CTT GTG TTC AGG AGC CAG CAG GGA GCT GGG AGC-3′. The DNA scaffold contains 14 nucleotides of upstream DNA, 23 nucleotides of downstream DNA and a mismatch bubble of 11 nucleotides, of which 9 nucleotides of the template DNA base-pairs with the RNA. The DNA oligos were synthesized by IDT and dissolved in water.

RNA and template DNA were mixed in equimolar ratio (30 μM) in 20 mM HEPES-NaOH, pH 7.5, 100 mM NaCl, and 3 mM $MgCl_2$, and annealed by heating up at 60 °C for 4 min followed by decreasing the temperature by 1 °C min$^{-1}$ steps to a final temperature of 30 °C in a thermocycler. *S. scrofa* Pol II (50 pmol) and the RNA–template DNA hybrid (100 pmol) were incubated for 10 min at 30 °C, followed by addition of the non-template DNA (200 pmol) and incubation at 30 °C for another 10 min. After adding DSIF and SPT6 (150 pmol each), phosphorylation reaction was carried out with 0.4 μM P-TEFb and 1 mM ATP in SEC100 buffer for 30 min at 30 °C. The assembled EC-DSIF-SPT6 complex was applied onto a Superose 6 Increase 3.2/300 column (GE Healthcare) equilibrated with SEC100 buffer containing 0.5 mM tris(2-carboxyethyl)phosphine (TCEP) instead of DTT. U1 snRNP was further purified on the Superose 6 Increase 3.2/300 column (GE Healthcare) in SEC100 buffer with 0.5 mM TCEP. The peak fraction of the EC-DSIF-SPT6 complex was mixed with the peak fraction of U1 snRNP at a molar ratio of 1:1.5, incubated on ice for 30 min, crosslinked with 0.05% glutaraldehyde on ice for 45 min, and used directly for freezing grids. Similar results were obtained when combining the EC-DSIF-SPT6 complex and U1 snRNP before the size exclusion purification step.

Samples were diluted to a concentration of ~150 nM, and 2 μl was applied to each side of the R2/2 UltrAuFoil grids (Quantifoil) that had been glow-discharged for 100 s. After incubation of 10 s and blotting for 4 s, the grid was vitrified by plunging it into liquid ethane with a Vitrobot Mark IV (Thermo Fisher) operated at 4 °C and 100% humidity.

## Cryo-EM data collection and processing
Cryo-EM data were collected on the 300 kV Titan Krios (Thermo Fisher) with a K3 summit direct detector (Gatan) and a GIF quantum energy filter (Gatan) operated with a slit width of 20 eV. Automated data collection was performed with SerialEM[47] at a nominal magnification of 81,000x, corresponding to a pixel size of 1.05 Å/pixel. Image stacks of 40 movie frames were collected with a defocus range of -0.5 to -2.0 μm in electron counting mode and a dose rate of 1.01 e⁻/Å²/frame. A total of 18,830 image stacks were collected.

All movie frames were aligned, and the contrast transfer function (CTF) parameters were calculated in Warp[48]. Particles in 380 pixels x 380 pixels were selected by automatic particle picking in Warp. The following steps were performed in RELION 5.0 (refs. [49],[50].) to exclude bad particles from the dataset: 1) Two-dimensional (2D) classification was performed, and particles in bad classes with poorly recognizable features were excluded. 2) In the second round of 2D classification, free U1 snRNP particles were excluded, leaving only Pol II containing particles. 3) The remaining particles were refined using three-dimensional (3D) refinement with a soft mask on Pol II and divided into six classes using 3D classification in RELION with local fine-angle search (0.9°). All 3D classes with bad particles were discarded.

All good particles with Pol II were combined and 3D refined using a soft mask on Pol II. To separate Pol II alone particles from the U1 snRNP containing particles, signal subtraction followed by focused 3D classification of the subtracted particles without alignment was performed using a large mask near the RNA exit site. Classes with an extra density corresponding to U1 snRNP were combined (12.4%), reverted to original particles, and 3D refined with a soft mask on Pol II. Subsequently, the SPT6-containing particles were selected using particle subtraction followed by 3D-classifcation without alignment with a soft mask on SPT6. Particles containing SPT6 were combined (20.6%), reverted to original particles, and 3D refined with a soft mask covering EC-DSIF-SPT6-U1 snRNP. This resulted in the final overall map of EC-DSIF-SPT6-U1 snRNP with 52,065 particles at 3.5 Å. To improve the local resolution of SPT6 and U1 snRNP, soft masks were applied individually onto SPT6$^{Core}$ and U1 snRNP. Following particle subtraction in RELION, particles were imported into CryoSPARC[51] for local refinement followed by Global CTF refinement and local refinement, resulting in a 6.2 Å map for SPT6$^{Core}$ and a 7.3 Å map for U1 snRNP.

## Model building and refinement
Initial models of Pol II (PDB: 7B0Y)[13] and DSIF (PDB: 9HVQ)[28] were rigid-body fitted into the overall map in Chimera[52]. Initial model of U1 snRNP (PDB: 7B0Y)[13] was rigid-body fitted into the focused refined map of U1 snRNP, while the model of SPT6 (PDB: 9HVQ)[28] was fitted into the focused refined map of SPT6. The Pol II model was manually adjusted in Coot[53]. The resulting complete model of EC-DSIF-SPT6-U1 snRNP was then real-space refined in the locally filtered and sharpened overall map using structure restraints of U1 snRNP, DSIF, and SPT6 in PHENIX[54].

Figures were generated using PyMOL (The PyMOL Molecular Graphics System, Version 2.0 Schrödinger, LLC.) and Chimera X[55].

## Reporting summary
Further information on research design is available in the Nature Portfolio Reporting Summary linked to this article.

## Data availability
The cryo-EM reconstructions and final model were deposited with the EMDB under accession codes EMD-53087 (overall map), EMD-53088 (focused refined map of SPT6), EMD-53089 (focused refined map of U1 snRNP), and the PDB under accession code 9QEQ. Source data are provided with this paper.

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

## Acknowledgements

We thank all members of the Zhang lab for discussion. We thank C. Dienemann and U. Steuerwald for support at the microscope and T. Schulz for the pig thymus. We thank the LMB scientific computing for maintaining the computing cluster and K. Turton for maintaining the insect cell facility. P.C. is supported by the Max Planck Society. S.Z. is funded by the Medical Research Council, as part of United Kingdom Research and Innovation (also known as UK Research and Innovation) with the MRC file reference number MC_UP_1201/30. For open access, the MRC Laboratory of Molecular Biology has applied a CC BY public copyright license to any Author Accepted Manuscript version arising.

## Author contributions

L.Z. purified proteins and performed biochemistry experiments. C.B. performed ITC experiments. S.A. assisted data processing. Y.G. purified proteins. K.Z., J.S., and K.M. generated the Pol II cell line. P.C. and S.Z. supervised the project. S.Z. designed and performed experiments, purified proteins, collected and analyzed cryo-EM data, and wrote the manuscript with input from all other authors.

## Competing interests

The authors declare no competing interests.
