## [Transparent Peer Review file · Nature Communications]

Structure of a transcribing Pol II-DSIF-SPT6-U1 snRNP complex

Corresponding Author: Dr Suyang Zhang

Version 0:

Reviewer comments:

Reviewer #1

(Remarks to the Author)

Zhang et al. present a cryo-EM-based structural analysis of a Pol II-DSIF-SPT6-U1 snRNP complex and interaction studies, through which they identify multiple contacts between U1 snRNP and Pol II-SPT6. Their pull-down experiments suggest a more stable Pol II-U1 snRNP interaction upon P-TEFb-mediated Pol II CTD phosphorylation and further stabilization in the presence of SPT6. Additional pull-down experiments confirm that the interaction involves contacts of phosphorylated CTD to the RRM domain of U1-70K protein. Pull-down studies and ITC assays with diverse SPT6 truncation variants reveal a direct interaction of SPT6 with the U1A protein via a C-terminal region of SPT6, consistent with a high-confidence AlphaFold3 prediction of an SPT6-U1A interaction.

The observations reported here agree with a previous publication that revealed direct contacts between U1-70K protein and Pol II subunits RPB2 and RPB12, and document additional contacts of U1 snRNP with phosphorylated Pol II CTD and SPT6. While SPT6 and U1 snRNP are tethered to Pol II in a highly flexible manner, leading to low local resolution of the corresponding regions in the cryo-EM reconstruction and failure to resolve U1 snRNP contacts to phosphorylated CTD or SPT6 directly, the newly described interactions are supported by systematic interaction studies and can be reconciled in light of the model that the authors present. The results, thus, are of potential interest to researchers concerned with the molecular mechanisms underlying the regulation of eukaryotic gene expression, and the physical and functional coupling of gene expression processes. The manuscript is written clearly and very concisely, and should be accessible to a broad audience. The work conducted appears technically sound.

Specific comments:

1. The U1 snRNP region is not well defined in any of the maps. Compared to U1 snRNP in other contexts (e.g., as part of the spliceosomal pre-A complex, PDB 7VPX), there are substantial conformational differences to U1 snRNP as modeled in the Pol II-DSIF-SPT6-U1 snRNP complex. E.g., the U1 snRNA-5'-splice site duplex in the present model is positioned quite differently relative to the remainder of U1 snRNP. Furthermore, the U1C protein, which consolidates this duplex, has seemingly not been modeled. The authors should very judiciously describe differences in their model to previous U1 snRNPs models and how well these differences are supported by their cryo-EM reconstructions.
2. Furthermore, the authors should discuss possible functional consequences of obvious or apparent differences in the U1 snRNP conformation (and possibly composition?) in the present complex compared to U1 snRNP in other contexts. For instance, is the positioning of U1 snRNP in the present complex compatible with subsequent formation of a spliceosomal pre-A complex?
3. "U1 snRNP bound directly to Pol II and its binding increased with Pol II phosphorylation by P-TEFb" (line 55) and "association of U1 snRNP to phosphorylated Pol II is further enhanced by SPT6" (line 66) - As pull-downs were done in triplicates, the authors should quantify the amount of U1 snRNP (represented, e.g., by U1-70K) relative to Pol II (represented, e.g., by RPB2) in elution fractions to clearly demonstrate (significance analysis) phosphorylation-dependent increase in the interaction and further enhancement in the presence of SPT6.
4. Line 129: "This multivalent interaction facilitates the swift recruitment of U1 snRNP to the nascent transcripts, thereby allowing efficient and accurate co-transcriptional spliceosome assembly" – This statement should be tuned down. While

several interaction sites of U1 snRNP to DSIF/SPT6-modified Pol II have been demonstrated convincingly, the importance of these contacts for co-transcriptional spliceosome assembly has not been tested. Furthermore, please refer to point 2 above – in which way is the present model compatible with subsequent steps of spliceosome assembly?

5. Line 133: “Therefore, the observed direct interactions of Pol II and SPT6 with U1 snRNP likely also ensure transcription processivity during elongation” – The authors should briefly discuss how U1 snRNP might enhance transcription processivity based on the interactions reported here. How does (or could) U1 snRNP modulate Pol II activity?

Reviewer #2

(Remarks to the Author)

In this study, the authors investigate co-transcriptional splicing by resolving the cryo-EM structure of a Pol II-DSIF-SPT6-U1 snRNP complex. This represents a more complete structure of a plausible U1-containing Pol II elongation complex compared with the previously published Pol II-U1 snRNP structure. The authors further identify and test an AlphaFold 3 prediction of the U1 snRNP subunit U1A with SPT6, which has not previously been associated with splicing, and show one pulldown indicating a preferred interaction of phosphorylated Pol II with the U1 snRNP. The manuscript is well structured and easy to follow. Although the findings are interesting, they provide minor new insights. The experimental cryo-EM structure shows to our knowledge no new details beyond published Pol II-elongation factor or Pol II-U1 snRNP complexes and the cellular relevance of even the formerly reported U1-Pol II interface remains untested. If the authors would like to increase the impact of their work, they may wish to consider any form of probing of U1-Pol II or SPT6-U1A interactions in any functional assay (in cells or in functional in vitro assays such as transcription, splicing). Owing to the associated efforts, we leave the decision on this to the authors.

Comments:

1. In Figure 1, it would be great to add additional input lanes for Pol II and Spt4/5/6 (also to see background binding to the beads) to see which bands correspond to the individual proteins. The authors may also wish to move the gel of their cryo-EM sample (ED 2b) to the main text, to show the stoichiometric binding of DSIF in the presence of nucleic acids, matching the structure of the EC-DSIF-SPT6-U1 snRNP complex shown in Figure 1c.

2. The interaction of U1 snRNP and SPT6 is tested using different SPT6 truncation constructs. They show that the SPT6 CTR binds to U1A and is sufficient to mediate their in vitro interaction. In Figure 2b, please indicate the truncated construct in the double band. Owing to the precise nature of the AF3 predictions, the authors should also minimally test this prediction more precisely, by mutating conserved residues in the interface, for example through the mutation of either Trp1681 (SPT6 CTR) or Tyr31 (U1A).

3. In Extended Data Figure 1, the pulldown shows binding to the phosphorylated CTD that interacts with the U1-70K RRM, but the background binding of, most importantly, the U1-70K RRM domain is not tested. Please assess background binding of U1-70K RRM to the beads and compare U1-70K RRM binding to phosphorylated and unphosphorylated CTD constructs.

4. In the ITC measurements in Extended Data Fig. 5 and also based on the pulldown in Fig. 2e (lane 6), it looks as if the NTR also contributes to U1 binding. Can the authors please comment on this in the manuscript and the impact on their model?

Version 1:

Reviewer comments:

Reviewer #1

(Remarks to the Author)

In the revised version of their manuscript, the authors have adequately addressed the points raised by this reviewer.

Reviewer #2

(Remarks to the Author)

The authors have addressed all in vitro concerns and we are happy to support publication. We wish to encourage the authors in future work to validate the interfaces of the 2021 U1-Pol II structure and those reported here in cells. The field will greatly benefit from knowing whether the reported interactions are relevant for gene expression in vivo.

Detailed list of responses to Reviewer's comments

Zhang et al. Structure of a transcribing Pol II-DSIF-SPT6-U1 snRNP complex

NCOMMS-25-20107-T

Reviewer #1 (Remarks to the Author):

Zhang et al. present a cryo-EM-based structural analysis of a Pol II-DSIF-SPT6-U1 snRNP complex and interaction studies, through which they identify multiple contacts between U1 snRNP and Pol II-SPT6. Their pulldown experiments suggest a more stable Pol II-U1 snRNP interaction upon P-TEFb-mediated Pol II CTD phosphorylation and further stabilization in the presence of SPT6. Additional pulldown experiments confirm that the interaction involves contacts of phosphorylated CTD to the RRM domain of U1-70K protein. Pulldown studies and ITC assays with diverse SPT6 truncation variants reveal a direct interaction of SPT6 with the U1A protein via a C-terminal region of SPT6, consistent with a high-confidence AlphaFold3 prediction of an SPT6-U1A interaction.

The observations reported here agree with a previous publication that revealed direct contacts between U1-70K protein and Pol II subunits RPB2 and RPB12, and document additional contacts of U1 snRNP with phosphorylated Pol II CTD and SPT6. While SPT6 and U1 snRNP are tethered to Pol II in a highly flexible manner, leading to low local resolution of the corresponding regions in the cryo-EM reconstruction and failure to resolve U1 snRNP contacts to phosphorylated CTD or SPT6 directly, the newly described interactions are supported by systematic interaction studies and can be reconciled in light of the model that the authors present. The results, thus, are of potential interest to researchers concerned with the molecular mechanisms underlying the regulation of eukaryotic gene expression, and the physical and functional coupling of gene expression processes. The manuscript is written clearly and very concisely, and should be accessible to a broad audience. The work conducted appears technically sound.

We thank this reviewer for the positive comments and constructive suggestions.

Specific comments:

1. The U1 snRNP region is not well defined in any of the maps. Compared to U1 snRNP in other contexts (e.g., as part of the spliceosomal pre-A complex, PDB 7VPX), there are

substantial conformational differences to U1 snRNP as modeled in the Pol II-DSIF-SPT6-U1 snRNP complex. E.g., the U1 snRNA-5'-splice site duplex in the present model is positioned quite differently relative to the remainder of U1 snRNP. Furthermore, the U1C protein, which consolidates this duplex, has seemingly not been modeled. The authors should very judiciously describe differences in their model to previous U1 snRNPs models and how well these differences are supported by their cryo-EM reconstructions.

We thank the reviewer for this comment. Human U1 snRNP consists of four stem loops. Although the interaction of U1 snRNP proteins with their respective stem loops is very stable, the relative orientation of these stem loops to each other is rather flexible. We superimposed our U1 snRNP structure with U1 snRNP in the human pre-A complex (7VPX)¹ on U1-70K (Fig. R1a). As the reviewer pointed out, there are movements of the three other stem loops. On the other hand, different motions could be observed when superimposing the human pre-A complex U1 snRNP with the human U1 snRNP crystal structure (3PGW)², highlighting the mobile nature of U1 snRNP stem loops (Fig. R1b).

The position of the 5'SS-U1 snRNA duplex differs in all three structures (Fig. R1c). The cryo-EM map for 7VPX has only weak densities for U1 snRNP, with densities corresponding to U1-70K, the 5'SS-U1 snRNA, U1C and most stem loops missing (Fig. R1d). Therefore, it would be difficult to compare the different conformations of U1 snRNP between these structures, as they are likely derived from the intrinsic mobile nature of U1 snRNP stem loops or crystal packing. The U1 snRNP model that we used to fit into our focused refined map is derived from the cryo-EM structure of Pol II-U1 snRNP (7B0Y)³ which has a local resolution of 5.9 Å for the entire U1 snRNP and 3.5 Å for U1-70K (Fig. R2).

U1C is only built in the pre-A complex structure (7VPX)¹, but absent in our and U1 snRNP crystal structures (3PGW)². We agree that previous studies have shown that U1C facilitates U1 snRNA-5'SS duplex formation. On the other hand, mass spectrometry studies showed that U1C is sub-stoichiometric or absent in some populations of human U1 snRNP⁴. Therefore, U1C is likely sub-stoichiometric in our purified U1 snRNP from HeLa cells. In addition, the movement of the U1 snRNA-5'SS duplex in our EC-DSIF-SPT6-U1 snRNP structure results in a larger gap between U1 snRNA-5'SS duplex and the Sm-ring, where U1C normally binds. This may also lead to destabilization of U1C. This conformation of the U1 snRNA-5'SS duplex is consistent with the previous Pol II-U1 snRNP structure (7B0Y)³. As we did not observe clear densities corresponding to U1C in any of our reconstructions, we did not include U1C in the model.

We included an additional Extended Data Fig. 4a, b to illustrate the U1 snRNP movements and associated text in Lines 95-99.

Fig. R1 | Comparison of the U1 snRNP structures, superimposed on U1-70K. a, Comparison of the U1 snRNP structure presented in this work (9QEQ, coloured as Fig. 2c) with that in the human pre-A complex (7VPX, in light grey). **b,** Comparison of the U1 snRNP structure in 7VPX (grey) with the human U1 snRNP crystal structure (3PGW, in blue). Arrows indicate the movement of stem loops. **c,** Comparison of 5' SS-U1 snRNA duplexes in this work (9QEQ), human pre-A complex (7VPX) and human U1 snRNP crystal structure (3PGW). U1 snRNA is colour as a and b, while the pre-mRNA is coloured in different shades of red. U1C is only built in 7VPX and coloured in yellow. **d,** Cryo-EM density of the human pre-A complex (EMD-32076) in light grey with the model 7VPX fitted (coloured as above).

Fig. R2 | Cryo-EM densities of the focused refined map for U1 snRNP from Zhang et al. 2021 Fig. S4³. Densities for the 5' SS-U1 snRNA duplex are shown in grey mesh with the RNA model fitted.

Extended Data Fig. 4 | Superposition with previous spliceosome structures. a, b, The U1 snRNP model of the EC-DSIF-SPT6-U1 snRNP (coloured as Fig. 2c) is superimposed onto the U1 snRNP crystal structure (PDB 3CW1 in white) on the Sm-ring, showing movements of stem loops with associated proteins. The 5' SS-U1 duplex (red and slate) rotates away from the Sm-ring, leaving a bigger gap at the binding site for U1C. 5' SS of the crystal structure is coloured orange.

2. Furthermore, the authors should discuss possible functional consequences of obvious or apparent differences in the U1 snRNP conformation (and possibly composition?) in the present complex compared to U1 snRNP in other contexts. For instance, is the positioning of U1 snRNP in the present complex compatible with subsequent formation of a spliceosomal pre-A complex?

We thank the reviewer for this comment. As discussed in point 1, we think the conformational differences are due to the mobile nature of U1 snRNP stem loops relative to each other, combined with the limited resolution of most existing U1 snRNP cryo-EM maps and crystal structures. It is therefore difficult to compare these structures to conclude any genuine conformational or compositional differences in splicing as an isolated process or a co-transcriptional process.

No densities connecting U1 and U2 snRNP was observed in the cryo-EM reconstruction of the human pre-A complex¹, making it difficult to interpret their relative orientations. Authors of this study have depicted the location of U1 snRNP as putative¹. We have therefore superimposed our EC-DSIF-SPT6-U1 snRNP structure with the yeast A complex⁵. The superposition reveals that U2 snRNP is accommodated on the surface of elongating Pol II, and therefore subsequent steps of spliceosome assembly likely can occur on Pol II (Extended Data Fig. 4c-d). We have added this discussion of subsequent spliceosome formation in Lines 147-151.

Extended Data Fig. 4 | Superposition with previous spliceosome structures. c, d, Superposition of the EC-DSIF-SPT6-U1 snRNP structure with the yeast pre-spliceosomal A complex (PDB: 6G90) on U1-70K indicates that pre-spliceosome formation may occur on the Pol II surface (grey surface). Yeast U1 snRNP is shown in white and U2 snRNP in light

green. The pre-mRNA of the EC-DSIF-SPT6-U1 snRNP complex is depicted in red, while that of the yeast pre-spliceosome in orange.

3. “U1 snRNP bound directly to Pol II and its binding increased with Pol II phosphorylation by P-TEFb” (line 55) and “association of U1 snRNP to phosphorylated Pol II is further enhanced by SPT6” (line 66) - As pulldowns were done in triplicates, the authors should quantify the amount of U1 snRNP (represented, e.g., by U1-70K) relative to Pol II (represented, e.g., by RPB2) in elution fractions to clearly demonstrate (significance analysis) phosphorylation-dependent increase in the interaction and further enhancement in the presence of SPT6.

We thank the reviewer for the suggestion. We have quantified the U1-70K bands in this pulldown and normalized the intensity to the RPB2 bands and against the Pol II+U1 snRNP condition, with error bars indicating standard deviation. Significance was calculated using unpaired Student’s t-test (indicated with asterisk). The increases of U1 snRNP binding upon Pol II phosphorylation and SPT6 binding are statistically significant. This is now included in Fig. 1c.

Fig. 1c | The amount of U1-70K in the elution was quantified and normalized to respective RPB2 band in each condition and against the Pol II+U1 snRNP condition, with error bars indicating standard deviation. Significance was calculated using unpaired Student’s t-test. * and ** indicate $P \leq 0.05$ and $P \leq 0.01$, respectively.

4. Line 129: “This multivalent interaction facilitates the swift recruitment of U1 snRNP to the nascent transcripts, thereby allowing efficient and accurate co-transcriptional spliceosome assembly” – This statement should be tuned down. While several interaction sites of U1

snRNP to DSIF/SPT6-modified Pol II have been demonstrated convincingly, the importance of these contacts for co-transcriptional spliceosome assembly has not been tested.

Furthermore, please refer to point 2 above – in which way is the present model compatible with subsequent steps of spliceosome assembly?

We thank the reviewer for raising this point and have modified our sentences accordingly in Lines 16-18 and Lines 169-172. As discussed in point 2, superposition of the yeast pre-spliceosome structure⁵ onto the EC-DSIF-SPT6-U1 snRNP complex reveals that U2 snRNP is accommodated on the elongating Pol II, indicating that the pre-spliceosome may assemble co-transcriptionally.

5. Line 133: “Therefore, the observed direct interactions of Pol II and SPT6 with U1 snRNP likely also ensure transcription processivity during elongation” – The authors should briefly discuss how U1 snRNP might enhance transcription processivity based on the interactions reported here. How does (or could) U1 snRNP modulate Pol II activity?

We thank the reviewer for raising this point and have expanded on how U1 snRNP modulates Pol II activity in the Discussion (Lines 152-168). Previous studies using U1 snRNA antisense morpholino oligonucleotides and transcriptome profiling from human cells revealed that U1 snRNP recruitment to the nascent transcripts is required to suppress premature cleavage and polyadenylation from cryptic polyadenylation signals^{6, 7, 8}. A recent work from the Adelman lab found that U1 snRNP stimulates synthesis of long introns by increasing Pol II elongation rate in mammalian cells⁹. In the absence of U1 snRNP, Pol II is more susceptible to termination and arrest, preventing the production of full-length transcripts.

A possible explanation is that U1 snRNP may prevent the recruitment or stable engagement of the 3' end processing machinery to cryptic sites. Furthermore, the integrator complex can lead to pre-mature termination and cleavage of transcripts in the promoter-proximal region. Superposition of the integrator-containing pre-termination complex with the EC-DSIF-SPT6-U1 snRNP structure reveals that the integrator tail module, which is required to facilitate the release of nucleic acids from the Pol II cleft and transcription termination, clashes with U1 snRNP on the Pol II surface (Extended Data Fig. 6). Furthermore, the integrator core module clashes with the SPT6 core domain. Therefore, U1 snRNP may prevent stable binding of the integrator complex while stabilizing SPT6 on Pol II. Additionally, U1 snRNP may compete

with the integrator-associated protein phosphatase 2A for the Pol II CTD to prevent its dephosphorylation and transcription termination.

Extended Data Fig. 6 | The integrator tail module clashes with U1 snRNP on Pol II. a, Superposition of the integrator-containing pre-termination complex (PDB: 8RBX) with the EC-DSIF-SPT6-U1 snRNP structure reveals that the integrator tail module (INTS10-13-14-15, in cyan) clashes with U1 snRNP on Pol II. **b,** Close-up view that shows the clashes between the integrator tail module (cyan) and the Sm-ring (pink) and U1 snRNA (slate), while the SPT6 core (blue) clashes with the integrator core module (white).

Reviewer #2 (Remarks to the Author):

In this study, the authors investigate co-transcriptional splicing by resolving the cryo-EM structure of a Pol II-DSIF-SPT6-U1 snRNP complex. This represents a more complete structure of a plausible U1-containing Pol II elongation complex compared with the previously published Pol II-U1 snRNP structure. The authors further identify and test an AlphaFold 3 prediction of the U1 snRNP subunit U1A with SPT6, which has not previously been associated with splicing, and show one pulldown indicating a preferred interaction of phosphorylated Pol II with the U1 snRNP. The manuscript is well structured and easy to follow. Although the findings are interesting, they provide minor new insights. The experimental cryo-EM structure shows to our knowledge no new details beyond published

Pol II-elongation factor or Pol II-U1 snRNP complexes and the cellular relevance of even the formerly reported U1-Pol II interface remains untested. If the authors would like to increase the impact of their work, they may wish to consider any form of probing of U1-Pol II or SPT6-U1A interactions in any functional assay (in cells or in functional in vitro assays such as transcription, splicing). Owing to the associated efforts, we leave the decision on this to the authors.

We thank this reviewer for the feedback and constructive comments.

We agree with the reviewer that assessing the effect of disrupting the Pol II-U1 snRNP or SPT6-U1A interface on co-transcriptional splicing would be very interesting. However, as the reviewer mentioned, this involves substantial efforts which will require generating new cell lines, using sophisticated genome-wide sequencing of nascent RNA, and/or setting up co-transcriptional splicing assays in extracts. Although these are clearly interesting lines of research for the future, they are beyond the scope of this manuscript. It is also unclear whether these interface mutations are viable and how they would affect other cellular processes, possibly rendering such analysis very difficult.

We have previously assessed the effect of U1 snRNP on transcription elongation in vitro. The speed of Pol II elongation in vitro was not affected by U1 snRNP³, thereby allowing co-transcriptional recruitment of U1 snRNP without disrupting transcription elongation.

Comments:

1. In Figure 1, it would be great to add additional input lanes for Pol II and Spt4/5/6 (also to see background binding to the beads) to see which bands correspond to the individual proteins. The authors may also wish to move the gel of their cryo-EM sample (ED 2b) to the main text, to show the stoichiometric binding of DSIF in the presence of nucleic acids, matching the structure of the EC-DSIF-SPT6-U1 snRNP complex shown in Figure 1c.

We thank the reviewer for raising this point. We have added additional input lanes for non-phosphorylated Pol II, phosphorylated Pol II, and DSIF+SPT6 in Fig. 1b. We also included a negative control to assess the non-specific binding of DSIF+SPT6 to the resin. No background binding of DSIF+SPT6 to the Strep-Tactin resin was observed as shown in Fig. 1b lane 8.

We have now split Fig. 1 into two figures for better presentation and moved the previous Extended Data Fig. 2b to Fig. 2b to show the stoichiometry of the cryo-EM sample next to the structure.

Fig. 1b | Pulldown of U1 snRNP using TwinStrep-tagged human Pol II in different phosphorylation states, along with elongation factors DSIF and SPT6. The prey protein U1-70K as a part of U1 snRNP is highlighted in a purple rectangle. The pulldown was performed in the absence of any nucleic acids and repeated in triplicates. pPol II: phosphorylated Pol II, pRPB1: phosphorylated RPB1. Asterisk indicates RPB1 degradation.

2. The interaction of U1 snRNP and SPT6 is tested using different SPT6 truncation constructs. They show that the SPT6 CTR binds to U1A and is sufficient to mediate their *in vitro* interaction. In Figure 2b, please indicate the truncated construct in the double band. Owing to the precise nature of the AF3 predictions, the authors should also minimally test this prediction more precisely, by mutating conserved residues in the interface, for example through the mutation of either Trp1681 (SPT6 CTR) or Tyr31 (U1A).

We thank the reviewer for these suggestions. The doublet band observed in Fig. 2b (now Fig. 3b lanes 3, 5, 9) for SPT6 is due to a gel-running artefact that occasionally occurs for SPT6 and not a protein truncation. The same batch of SPT6 protein was used in experiments for Fig. 3c (lanes 3 and 4) and Fig. 3e (lanes 3, 4, 9 and 10), where no doublet bands were observed.

We attempted to disrupt the predicted SPT6^{CTR}-U1A interaction by either replacing the SPT6 C-terminal helix with GSA linkers that can span the length of the helix (SPT6^{Δhelix}) or mutating large aromatic and hydrophobic residues in the SPT6 C-terminal helix to alanines (SPT6^{3A}: W1674A, W1681A, L1682A). Pulldown of U1A using different MBP-tagged SPT6 constructs revealed that SPT6^{Δhelix} abolished U1A binding to the same extent as SPT6^{ΔCTR}, while SPT6^{3A} strongly reduced U1A binding, validating the predicted interaction of SPT6^{CTR}-U1A. We have now included this result in Fig. 4b and a description of the result in Lines 137-141.

Fig. 4b | Mutations that disrupt the predicted SPT6^{CTR}-U1A interaction strongly reduced the amount of U1A bound to MBP-tagged SPT6. The SPT6 C-terminal helix was replaced with GSA linkers that can span the length of the helix (SPT6^{Δhelix}) or large aromatic and hydrophobic residues in the SPT6 C-terminal helix were mutated to alanines (SPT6^{3A}: W1674A, W1681A, L1682A). This pulldown was repeated in triplicates.

3. In Extended Data Figure 1, the pulldown shows binding to the phosphorylated CTD that interacts with the U1-70K RRM, but the background binding of, most importantly, the U1-70K RRM domain is not tested. Please assess background binding of U1-70K RRM to the beads and compare U1-70K RRM binding to phosphorylated and unphosphorylated CTD constructs.

We thank the reviewer for raising this point. We have now assessed the binding of U1-70K^{RRM} to the amylose resin and to both phosphorylated and non-phosphorylated Pol II CTD. The pulldown assay showed no background binding of U1-70K^{RRM} to the amylose resin and U1-70K^{RRM} interacted specifically with the P-TEFb phosphorylated Pol II CTD, but not with the non-phosphorylated Pol II CTD. We have included this result in Fig. 1e.

Fig. 1e, Pulldown of U1-70K^{RRM} with MBP-tagged human Pol II CTD. U1-70K^{RRM} interacts specifically with the P-TEFb phosphorylated Pol II CTD (pCTD). P-TEFb consists of CDK9 and cyclin T1, a truncated cyclin T1 (1-272) was used. The prey protein U1-70K^{RRM} is highlighted in purple rectangle. This pulldown was repeated in triplicates.

4. In the ITC measurements in Extended Data Fig. 5 and also based on the pulldown in Fig. 2e (lane 6), it looks as if the NTR also contributes to U1 binding. Can the authors please comment on this in the manuscript and the impact on their model?

We thank the reviewer for raising this point. The NTD of SPT6 is very acidic, with a high number of glutamate and aspartate residues. The resulting strong negative charge of SPT6^{NTD} makes it a good mimic of the RNA substrate of the RRM domains. In the absence of the core domain in the SPT6^{NTD+CTR} construct, the NTD is brought into close proximity to the CTR which interacts specifically with U1A. This potentially allows the highly negatively charged NTD to non-specifically engage the positively charged RRM domains of U1A, resulting in an increased affinity for this construct. Although we cannot rule out the possibility that SPT6^{NTD} may contribute to U1 snRNP binding, the NTD itself is unable to bind to either U1A or U1 snRNP (Fig. 3c lane 8, 3e lane 12). In addition, deletion of the NTD did not affect the amount of U1A or U1 snRNP bound in comparison to wildtype SPT6 in the pulldown (Fig. 3c lane 10, 3e lane 14). Therefore, SPT6^{NTD} is insufficient and not necessary for the SPT6-U1A interaction. We have now added additional text in Lines 127-128 to describe this.

References

1. Zhang X, *et al.* Structural insights into branch site proofreading by human spliceosome. *Nat Struct Mol Biol* **31**, 835-845 (2024).

2. Weber G, Trowitzsch S, Kastner B, Luhrmann R, Wahl MC. Functional organization of the Sm core in the crystal structure of human U1 snRNP. *EMBO J* **29**, 4172-4184 (2010).
3. Zhang S, Aibara S, Vos SM, Agafonov DE, Luhrmann R, Cramer P. Structure of a transcribing RNA polymerase II-U1 snRNP complex. *Science* **371**, 305-309 (2021).
4. Hernandez H, *et al.* Isoforms of U1-70k control subunit dynamics in the human spliceosomal U1 snRNP. *PLoS One* **4**, e7202 (2009).
5. Plaschka C, Lin PC, Charenton C, Nagai K. Prespliceosome structure provides insights into spliceosome assembly and regulation. *Nature* **559**, 419-422 (2018).
6. Kaida D, *et al.* U1 snRNP protects pre-mRNAs from premature cleavage and polyadenylation. *Nature* **468**, 664-668 (2010).
7. So BR, *et al.* A Complex of U1 snRNP with Cleavage and Polyadenylation Factors Controls Telescripting, Regulating mRNA Transcription in Human Cells. *Mol Cell* **76**, 590-599 e594 (2019).
8. Berg MG, *et al.* U1 snRNP determines mRNA length and regulates isoform expression. *Cell* **150**, 53-64 (2012).
9. Mimoso CA, Adelman K. U1 snRNP increases RNA Pol II elongation rate to enable synthesis of long genes. *Mol Cell* **83**, 1264-1279 e1210 (2023).